# The Dual Role of Interferon Signaling in Myeloproliferative Neoplasms: Pathogenesis and Targeted Therapeutics

**DOI:** 10.3390/cancers17213480

**Published:** 2025-10-29

**Authors:** Valentina Bonuomo, Irene Dogliotti, Simona Masucci, Selene Grano, Arianna Savi, Antonio Frolli, Daniela Cilloni, Carmen Fava

**Affiliations:** 1Department of Clinical and Biological Sciences, Università degli Studi di Torino, 10124 Turin, Italydaniela.cilloni@unito.it (D.C.);; 2Division of Hematology 1 U, Department of Biotechnology and Health Sciences, Università degli Studi di Torino, 10124 Turin, Italy; 3Farmacia Clinica Onco-Ematologia, Azienda Ospedaliera Mauriziano di Torino, 10128 Turin, Italy; 4Department of Public Health and Pediatrics, University of Turin, 10126 Turin, Italy

**Keywords:** myeloproliferative neoplasms, interferon, signaling

## Abstract

**Simple Summary:**

Interferon families are a heterogeneous group of proteins playing a crucial regulatory role in immunomodulation and showing potent antitumor effects through a variety of different receptors. In this review we explore Interferon’s physiological roles, intracellular pathways and current therapeutic use in the field of myeloproliferative neoplasms.

**Abstract:**

Interferons (IFNs) are pleiotropic cytokines involved in antiviral defense, immune regulation, and tumor suppression. In myeloproliferative neoplasms (MPNs) and related disorders—including classical BCR, ABL1-negative MPNs, chronic myeloid leukemia (CML), and rarer entities such as chronic neutrophilic leukemia and hypereosinophilic syndromes—disease pathogenesis arises from a spectrum of somatic and structural genetic abnormalities and chronic inflammation, in which IFNs play a paradoxical role. They contribute to disease pathogenesis by promoting abnormal hematopoiesis and immune dysregulation, while also representing a therapeutic option capable of inducing hematologic and molecular remissions. This review outlines the biology and classification of IFNs, focusing on their signaling pathways and downstream effects in both normal and malignant hematopoiesis. We discuss the dual impact of IFN signaling on hematopoietic stem cells, including induction of proliferation, senescence, apoptosis, and DNA damage, and how these mechanisms may both sustain clonal evolution and facilitate disease control. Clinical data supporting the efficacy and safety of IFN-α, particularly pegylated formulations, in polycythemia vera, essential thrombocythemia, myelofibrosis, and chronic myeloid leukemia are reviewed, along with insights into next-generation IFNs and combination therapies. Understanding the dichotomous effects of IFNs in MPNs not only clarifies their role in disease biology but also informs their optimal use in clinical practice. This duality highlights the need for personalized approaches to IFN-based therapies.

## 1. Introduction

Interferons (IFNs) represent a distinct family of cytokines that play crucial physiological roles in immunomodulation. IFNs were initially identified by virologists, and owe their name to their antiviral properties, but subsequent work highlighted potent immunomodulatory effects and important roles in the antitumor immune response [1].

IFNs started to be employed as therapeutics in the field of viral infections, but later on were further developed as antitumoral agents in different settings [2], although unfortunately, their use in clinical practice is often limited by side effects [3].

Despite their essential regulatory functions, dysregulated IFN signaling—particularly involving type I IFNs—has been associated with autoimmune and inflammatory disorders, underscoring the risks of uncontrolled IFN activity [4,5].

Recently, a deeper understanding of IFNs’ mechanisms of action has renewed interest in their application as cancer therapeutics.

## 2. Materials and Methods

A systematic literature search was conducted on PubMed (National Library of Medicine) focusing on interferon signaling, pharmacology, and mechanisms of action in hematologic malignancies. 

To evaluate IFN efficacy in myeloproliferative settings, we applied the PICO framework, using ‘myeloproliferative neoplasms’ (including essential thrombocythemia, polycythemia vera, primary myelofibrosis, chronic myeloid leukemia, chronic neutrophilic leukemia, and the MDS/MPN overlap disorders chronic myelomonocytic leukemia and atypical chronic myeloid leukemia) as the condition, and ‘interferon’ as the intervention. The inclusion of MDS/MPN entities was justified by their biological and clinical overlap with classical MPNs and by the shared relevance of IFN-mediated signaling in these disorders. 

We included retrospective series, prospective studies, and randomized controlled studies where IFN monotherapy was administered. Studies with interferon combinations (e.g., together with JAK inhibitors) were excluded due to their infrequent use in clinical practice of western Countries, and to difficulty in discerning the contribution in patients’ outcome. Only articles in English were included. The resulting publications were manually screened by reading title and/or abstract and were considered separately as whether they referred to ET, PV, MF or other MPNs.

## 3. Interferon Families and Biological Functions

### 3.1. Families

The first interferon was discovered in 1957 by Isaacs and Lindenmann, followed by the identification of IFN-γ in 1965 and type III IFNs (λ) in 2003 [6,7,8].

Overall, there are 21 types of IFNs in humans (namely: 16 type-I, i.e., 12 IFNαs, IFNβ, IFNϵ, IFNκ, and IFNω; one type-II, IFNγ; and four type-III, IFNλ1, IFNλ2, IFNλ3, and IFNλ4); they were originally divided into three classes, type I, type II, and type III, based on their sensitivity to pH [9]. While all IFNs share antiviral and immunomodulatory properties through interaction with their specific receptors, they differ substantially in structure, chromosomal localization, and biological context of expression.

From a genomic standpoint, type I IFNs are encoded by a cluster of genes located on chromosome 9p21. In contrast, type II IFN (IFNγ) is encoded by a single gene situated on chromosome 12q15, while type III IFNs are organized as a compact gene cluster on chromosome 19q13 [10].

The type I IFN family has the broadest range of biological effects and is most widely studied in oncology; IFNα (encoded by 13 genes) and IFNβ (single gene) are the predominant subtypes [11]; IFNα, in particular, has been extensively studied for its anti-tumor activity against several malignancies [2]. Other members (ϵ, κ, ω) remain less characterized and show overlapping and species-specific activities [12,13].

Type II IFN or IFN-γ is structurally unrelated to the other two classes and is best known as a critical cytokine secreted during activated NK- and T-cell responses [7,14].

Finally, type III IFNs, or IFN-λs, are the least characterized IFN family, structurally related to type I IFNs and to the IL-10 family [13,15,16]. Among them, IFNλ4 exhibits particular biological properties: its expression is controlled by a polymorphic dinucleotide variant (ΔG/TT) in the IFNL4 gene, which affects protein production and modulates antiviral and inflammatory responses in a population-dependent manner [17].

These genomic and structural features account for the diversity of IFN-mediated signaling and help explain their distinct but sometimes overlapping roles in immunity and oncogenesis.

### 3.2. Signaling and Regulation

The various IFNs activities are mediated by binding to different heterodimeric receptors on recipient cell membranes (Figure 1): type-I IFNs signal via the IFNAR1 and IFNAR2 receptor complex, type-II through IFNGR1 and IFNGR2, and type-III using IFNlR1 and IL-10R2 receptors. Each receptor dimer is composed by a high affinity chain (IFNAR2, IFNGR1, IFNlR1) and a low-affinity one (IFNAR1, IFNGR2, IL10R2); all chains are specific for their cognate but IL-10R2 (which is shared with IL10, IL22 and IL26 pathways) [18,19,20,21].

The description of IFN receptor signaling provided herein mainly derives from studies performed in hematopoietic and immune cells, including macrophages, dendritic cells, and T-lymphocytes, as well as in epithelial and hepatocyte models. Although the JAK–STAT, MAPK, MTOR, and ULK1 pathways represent conserved mechanisms across IFN types, the strength of activation, stat heterodimer composition, and transcriptional output of interferon-stimulated genes (ISGs) vary depending on the cell type and its microenvironmental context [22,23,24].

Signaling through IFN receptors in turn activates Janus kinases (JAKs), which are shared for type I and III IFNs (JAK1 and TYK2), and distinct for type II (JAK1 and 2) [1,25]; high affinity IFN receptors, after phosphorylation by JAKs, recruit signal transducers and activators of transcription (STATs) (again, shared for type I and III, while partly different for type II), that migrate to the cell nucleus, form a complex and activate ISGs, whose protein products drive several distinct biological functions through JAK-STAT, MAP kinase, mTOR, and ULK1 pathways [26,27,28].

Responses to IFN-initiated signaling are further mediated by the IFN regulatory factor (IRF) family: the complex formed by STAT1 and STAT2 and IRF9 (called ISGF3) [29], works as transcriptional activator, promoting IFN-inducible genes by binding IFN-stimulated response elements (ISREs) in gene promoters, whereas IRF2 functions as a transcriptional attenuator, counterbalancing ISGF3 directly in the nucleus. Thus, in a mouse model lacking IRF2, there is a notable excess in type I IFN signaling with hyper-responsiveness to antigen stimulation [30].

Regarding IFNs genes transcription, it has been shown that type III IFN is highly dependent on NF-κB stimulation, while type I IFNs do not strictly depend on NF-κB signaling [31], as their production is stimulated in a similar fashion to other cytokines through pattern recognition receptors, such as Toll-like receptors.

Other than NF-κB, the IFN-γ gene also contains binding sites in its promoter region for several transcription factors, such as AP-1, CREB/ATF, NFAT, T-bet, Eomes and STATs [32,33]; indeed, stimulation via the T-cell receptor or through cytokines (IL-12, IL-18 but also type I IFN) induces IFN-γ gene expression thanks to the activation of these transcription factors.

Actually, it was also shown that effective IFN-γ signaling is contingent on a weak type I IFN signaling caused by a low constitutive production of type I IFNs [34].

#### Cell Specificity

Type-I and III IFNs are considered part of the innate immune system, with the primary role to activate antiviral states in cells, yet can also modulate adaptive immune responses. On the contrary, type-II IFN itself is a product of adaptive immunity, but can act on innate immunity cells such as macrophages.

It should also be noted that prolonged IFN signaling triggers regulatory feedback loops that prevent immune overactivation. These mechanisms can suppress the differentiation of dendritic cells, enhance the function of regulatory T cells, and increase the expression of immunosuppressive genes, ultimately fine-tuning the immune response.

Type I IFNs are produced predominantly by immune (dendritic plasmacytoid cells, monocytes, macrophages, and to a lesser extent NK and activated T cells) and stromal cells, although tumor cells are also capable of secreting them. They play a dual role by directly targeting malignant cells and indirectly enhancing antitumor immunity. This is achieved through the activation of cytotoxic T lymphocytes, B cells, and NK cells, while simultaneously inhibiting myeloid-derived suppressor cells and regulatory T cells [35,36].

Type II IFNs are synthesized by activated immune cells, such as NK cells and T cells. These cytokines are key mediators of antiviral defenses and immune regulation [7,14].

Type III IFNs, unlike their type I counterparts, are expressed in a more restricted manner, primarily at epithelial surfaces. Their expression is minimal in hematopoietic cells, reflecting their specialized role in epithelial tissue protection [6,31].

One major difference between type I and type III IFNs lies in their receptor distribution. While type I IFN receptors (IFNAR1 and IFNAR2) are ubiquitously present on all nucleated cells, type III receptors (IFNlR1) are largely confined to epithelial cells [37]. This localization underpins the unique function of type III IFNs in protecting epithelial barriers, such as those in the gut, lung, and liver [38]. For instance, type III IFNs are indispensable in managing norovirus infections, a role not adequately fulfilled by type I IFNs. Interestingly, in gut epithelial cells, type I IFN receptors are only found on the apical surface, leaving the basolateral surface devoid of them. This spatial arrangement emphasizes the importance of type III IFNs in orchestrating gut immune defenses in an intact tissue environment. While type III IFN signaling might appear weak in vitro, its efficacy is evident in vivo. Additionally, specific type I IFNs, such as IFNϵ and IFNk, contribute to the defense of the female reproductive tract and the skin, respectively [39,40,41,42].

## 4. Interferon and Inflammation

Patients with MPNs commonly exhibit chronic inflammation [43,44,45], characterized by the overexpression of various pro-inflammatory cytokines. Among these, G-CSF and GM-CSF primarily activate the JAK2/STAT5 signaling axis, promoting myelopoiesis, whereas IL-6 plays a dual role, acting through JAK2/STAT5 in hematopoietic progenitors and JAK1/STAT3 in immune and stromal cells [46]. This pleiotropic signaling contributes both to enhanced [47,48,49,50,51].

The severity of MPN-associated clinical symptoms—including fatigue, fever, night sweats, weight loss, and pruritus—as well as complications such as thrombosis (arterial and venous), splenomegaly, and bone marrow fibrosis, is closely correlated with systemic inflammation.

Interferon thus appears to have an ambivalent role in supporting chronic inflammation, although we are aware that it is used in the treatment of the myeloproliferative diseases debated in this review. while simultaneously serving as an effective treatment for MPNs. To shed light on this apparent paradox, identification of driver (JAK2, CALR, MPL) and non-driver (TET2, DNMT3A, SRSF2, SF3B1) mutations has prompted investigations into their relationship with the chronic inflammatory state observed in these disorders [52].

Inflammation in MPNs may result from two distinct mechanisms: (1) clonal inflammation, where the JAK2V617F mutation directly drives cytokine overproduction, or (2) pre-existing inflammation, which may create a pro-inflammatory microenvironment that predisposes to JAK/STAT-activating mutations.

Chronic inflammation is a well-established driver of malignancy, particularly in the context of persistent infections, metabolic disorders, and autoimmune diseases. In older individuals, a state of “clonal inflammation” may arise due to early genetic mutations (JAK2, TET2, DNMT3A, SRSF2, SF3B1), potentially fostering MPN development. Non-genetic inflammatory triggers—such as smoking, chronic inflammatory conditions, and metabolic dysregulation—may also contribute to MPN pathogenesis [53,54,55,56,57].

Many inflammatory cytokines overexpressed in MPNs are not directly induced by JAK2V617F or CALR mutations within the mutated cells themselves. Rather, mutant clones can trigger a broader pro-inflammatory response by influencing non-mutated immune cells in the microenvironment. For instance, JAK2V617F can upregulate IL-1β, IL-1Rα, and IP-10, which in turn stimulate further cytokine production by non-clonal monocytes and macrophages. In murine models expressing JAK2V617F, IL-1β knockout leads to reduced inflammation, megakaryopoiesis, myelofibrosis [49] and osteosclerosis [58,59], underscoring its pathogenic role. By contrast, CALR and MPL mutations are less directly associated with cytokine induction; in CALR-mutated MPNs, the inflammatory milieu is thought to be sustained primarily by non-mutated T cells [60,61].

Non-driver mutations (DNMT3A, TET2, SRSF2, SF3B1) also contribute to inflammation by activating NF-κB, a key regulator of inflammatory cytokines (IL-1β, TNF-α, TGF-β) and monocyte/macrophage function [62]. This inflammation may precede the acquisition of JAK2, CALR, or MPL mutations, supporting the hypothesis that a pro-inflammatory state may facilitate leukemogenesis [63,64].

IFN-α therapy, which represses IL-1β, NF-κB, and c-MET/HGF signaling, has demonstrated long-term remissions in both JAK2V617F- and CALR-mutated MPNs. Mutations in TET2, DNMT3A, ASXL1, and EZH2 have been linked to reduced efficacy of IFN-α therapy [65,66,67] although the underlying mechanisms remain unclear.

## 5. Rationale and Mechanisms of Action of Interferon in MPN

In the context of anti-cancer treatment setting, IFNs were mostly employed against hematological diseases, with MPNs patients showing the strongest responses; historically, IFNα was able to induce major responses in chronic myeloid leukemia (CML) and therefore has been a mainstay of treatment in this field before the approval of tyrosine kinase inhibitors (TKIs). Nevertheless, it remains a viable therapeutic option in BCR:ABL1 negative MPNs, especially in polycythemia vera (PV) and essential thrombocytosis (ET).

The clinical use of IFN-α in MPNs is associated with a characteristic spectrum of adverse effects. The most common include flu-like symptoms, fatigue, myalgia, and low-grade fever, which are usually transient and dose dependent. Hematologic toxicities, particularly leukopenia and thrombocytopenia, as well as mild hepatic enzyme elevations, may also occur. More rarely, autoimmune phenomena (e.g., autoimmune thyroiditis, lupus-like syndromes, psoriasis flares) and neuropsychiatric adverse effects, such as depression or cognitive changes, have been reported and may necessitate treatment interruption. Pegylated interferons display a more favorable tolerability profile, but careful clinical and laboratory monitoring remains essential during prolonged therapy [67,68,69].

In recent years, an emerging deeper understanding of the various mechanisms behind IFN’s unique effects on tumor biology has renewed interest in their application as cancer therapeutics in particular in MPN treatment. In the following sections we will further describe and analyze these mechanisms.

### 5.1. Exhaustion by Differentiation

IFNα has a distinct disease-modifying effect by acting on MPN Hematopoietic Stem Cells (HSCs) by inducing proliferation through IFNα/β receptor coupled to JAK1 and STAT1-dependent signaling [70]. Indeed, IFNα can push MPN HSCs carrying the most common JAK2V617F mutation into exhaustion by promoting entry into the cell cycle; moreover, several studies have shown similar effects in quiescent HSC populations leading to proliferation-associated functional exhaustion in wild-type (WT) mice [71].

In physiological conditions, transient type I IFN signaling activates dormant HSCs during infection or inflammation, allowing rapid hematopoietic recovery. However, sustained activation—such as during chronic treatment—causes long-term depletion of normal HSCs and may contribute to cytopenias and marrow suppression [70].

Mosca [72] employed a mathematical model showing that IFNα exhausts heterozygous and homozygous JAK2V617F HSCs by promoting their exit from quiescence and differentiation into progenitors. In this model, IFNα depletes homozygous JAK2V617F HSCs more effectively than heterozygous ones. Although its effect on CALR-mutated HSCs is weaker, IFNα preferentially targets type 2 CALR mutants over type 1.

These findings highlight the dual role of IFNα—as a therapeutic agent targeting malignant clones, but also as a stressor capable of inducing exhaustion in normal HSCs—underscoring the importance of dosing and treatment duration in clinical practice.

### 5.2. Senescence and Apoptosis

Prior studies in MPN mouse models, predominantly manifesting PV phenotypes, have demonstrated that IFNα can directly target JAK2V617F HSCs through pro-apoptotic mechanisms, in addition to proliferation-associated exhaustion [73,74,75]. 

Tong et al. [76], observed this pro-apoptotic effect in heterozygous-JAK2 mutant HSCs in individuals with ET treated with IFNα, while cell cycling was only modestly enhanced. Interestingly, homozygous cells seemed to re-enter quiescence through restoration of the TSC-mTOR signaling pathway or TP53 activation, implying that these mutant, quiescent cells are preserved and serve as residual disease-initiating stem cells. Molecular remission can be achieved by IFNα treatment [77]; however, rapid molecular relapse occurs in some individuals after IFNα discontinuation [78].

Relapsing cells might originate from quiescent mutant HSCs in individuals with ET. Transient, low-dose IFN stimulation promotes proliferation and differentiation of JAK2V617F+ megakaryocyte (Mk) primed HSCs during disease onset, whereas, upon treatment (including a chronic, therapeutic dose of IFNα), the mutant Mk-primed HSC population was reduced by promoting apoptosis or quiescence of mutant cells.

In this setting, quiescence serves as a double-edged sword: while its exit is necessary for therapeutic exhaustion, its reestablishment in selected clones represents a key mechanism of disease persistence and relapse after therapy withdrawal.

Overall, these findings underscore that IFNα effects on HSCs are context-dependent, influenced by mutation burden, signaling feedback loops, and treatment duration, which collectively determine whether IFNα drives exhaustion, apoptosis, or long-term quiescence.

### 5.3. Accumulation of DNA Damage and Reactive Oxygen Species

Prolonged stimulation by IFN-α leads to the accumulation of DNA damage and reactive oxygen species (ROS), ultimately resulting in the attrition of long-term hematopoietic stem cells (LT-HSCs) [79].

This concept was functionally explored by Austin et al. [73], whose findings suggest an increased sensitivity or primed state of Jak2+ mutated HSCs (Jak2+/VF LT-HSCs) to IFN-α-induced exhaustion. At the transcriptional level, chronic IFN-α treatment led to baseline enrichment of STAT1 target genes, increased expression of genes associated with DNA damage and mitochondrial activation, and sustained ROS production in Jak2+/VF HSCs compared with WT-HSCs, consistent with previous findings [80,81].

Notably, while acute IFN-α treatment induces cell cycle activation, DNA damage, and ROS production in both WT and Jak2+/VF HSCs and progenitors, chronic peg-IFN-α treatment is required to preferentially reduce quiescence, increase G1-phase accumulation, and enhance ROS production and DNA damage in Jak2+/VF LT-HSCs compared with WT LT-HSCs. These findings align with previously published data using unmodified IFN-α [75].

Recurrent activation of IFN signaling, as observed with repeated poly I:C exposure, can attenuate the HSC cycle response [82]. A once-weekly pegylated IFN-α regimen may mitigate this effect by allowing sufficient recovery time between doses.

Because chronic activation of LT-HSCs imposes sustained proliferative pressure, our findings indicate that replication stress acts as the detrimental mechanism leading to the depletion and functional decline of Jak2^V617F LT-HSCs. This interpretation is consistent with prior studies implicating replication stress as a driver of stem cell exhaustion [83], as well as with reports of increased replication fork stalling in JAK2V617F cells [84].

### 5.4. Anticlonogenic Effect

Recent work identified the PKCδ-ULK1-p38 MAPK signaling cascade as critical to IFNα-induced anti-neoplastic effects in MPN patients. In this pathway, PKCδ activation by IFN treatment mediates phosphorylation of ULK1 at Ser341 and Ser495, thereby activating ULK1. ULK1 then phosphorylates p38 MAPK, driving expression of ISGs to suppress tumor activity [85]. In primary MPN cells, the baseline pre-treatment expression of ULK1 and p38 MAPK was found to correlate with patient response to pegylated-IFNα treatment [86]. Importantly, the antineoplastic effects of IFN were found to be negatively regulated by IFN-induced caspase-mediated activation of Rho-associated protein kinases (ROCK1/2) [87]. ROCK1/2 expression is increased in MPN patients, and inhibition of these proteins potentiates the anticlonogenic effect of IFN treatment on primary MPN cells ex vivo.

### 5.5. Other Mechanisms

Interferon alpha-2 treatment reduces circulating neutrophil extracellular trap (NETs) levels that may play a pathogenic role in the thrombosis in myeloproliferative neoplasms [88].

## 6. Interferon Pharmacology

### 6.1. Pharmacodynamic Properties

Interferons differ not only in molecular structure but also in receptor distribution and downstream signaling kinetics, which determine their pharmacodynamic (PD) behavior.

All IFNs share an α-helical tertiary structure, yet with distinct topologies: type I and III IFNs form monomeric, four-helix bundles, while type II (IFN-γ) adopts a dimeric “swapped” E–F interface [89].

Each IFN consists of six secondary structural elements, designated A–F, of which helices A, C, D and F form an antiparallel four-helix bundle.

Type I IFNs, the largest family, include 13 functional IFN-α subtypes and single representatives for IFN-β, IFN-ε, IFN-κ, and IFN-ω. Although they signal through the same receptor complex (IFNAR1/IFNAR2), the various α-subtypes differ in receptor affinity and signaling potency, leading to distinct biological and therapeutic profiles [90]. These IFNs range in length from 165 to 166 amino acids and in molecular weight from 17.5 to 26 kDa.

The molecular weight of IFN-β varies depending on the form: IFN-β1a ~22.5 kDa and IFN-β1b ~18.5 kDa. The difference in molecular weight is due to glycosylation in IFN-β1a, which is recognized as having a complex effect on a wide range of molecular properties and functions [91].

In contrast to type I IFNs, type III IFNs (λ) are composed of shorter helices containing multiple kinks that form a more compact bundle and adopt structures more similar to the cytokine IL-22 than to type I IFNs. Unlike the monomeric IFNs type I and III, IFNγ forms a swapped dimer (E–F). The structure of IFNγ is most similar to that of IL-10, the founding member of the IL-10 family [13,92]. This confirms that each IFN family has a distinct α-helical scaffold that needs to be ‘handled’ by varying degrees of sequence variation to regulate binding to their cellular receptors.

IFN-γ is a 143-amino-acid polypeptide chain linked to 2 binding monomers. It has a molecular weight of 17–20 kDa, subject to slight variation due to glycosylation.

Pegylated interferon (PEG-IFN) consists of interferon alpha or gamma covalently linked to a polyethylene glycol (PEG) chain. PEG is a polymer capable of modifying the properties of the protein without significantly altering its biological function.

Pegylation prevents interferon from being rapidly eliminated from the body through metabolism and renal excretion. PEG makes the protein larger and more difficult for immune system cells to recognize, thus slowing the elimination process. This results in a longer half-life than non-pegylated interferon, allowing less frequent dosing, usually weekly or monthly instead of daily [93].

PEG-IFN molecular size varies depending on the PEGylation process and the size of the attached PEG molecule. The molecular weight for PEG-IFNα-2a is 170 kDA: ~19 kDA for the interferon component and ~150 kDA for the PEG chains. 156 kDA for PEG-IFNα-2b: ~19 kDA for the interferon component and ~150 kDA for the PEG chains.

### 6.2. Pharmacokinetic Properties

Native interferons are well absorbed when administered intramuscularly or subcutaneously. These routes exhibit protracted absorption, reaching peak levels in the serum or plasma after 1–8 h, followed by measurable concentrations for 4–24 h after injection for both interferons-α and γ. The peak serum concentrations following these routes are at least an order of magnitude lower than those achieved after an equivalent dose administered intravenously [94,95].

After intravenous administration, serum concentration peaked by the end of the 30 min infusion, then declined at a slightly more rapid rate than after intramuscular or subcutaneous drug administration, becoming undetectable 4 h after the infusion [96].

The bioavailability of interferons after subcutaneous injection is typically high, usually in the range of 80–90%. The volume of distribution is similar for both IFNα and γ ranging from 12 to 40 L, suggesting distribution mainly within extracellular fluids [97]. The half-life of this drug is approximately 3–8 h (intravenous administration) or 6–7 h (intramuscular and subcutaneous administration). Interferons are subject to little hepatic metabolism.

Metabolism occurs primarily through proteolytic degradation in renal tubular cells and the reticuloendothelial system, with minimal hepatic metabolism. In patients with renal impairment, clearance of non-pegylated IFNs may increase due to altered filtration and catabolism dynamics, whereas PEGylation decreases clearance, leading to sustained serum levels and prolonged half-life (up to 160 h for PEG-IFNα-2a and 120 h for ropeginterferon alfa-2b).

After a single subcutaneous injection of PEG-IFN serum concentrations of IFNα-2a are measurable within 3 to 6 h. Within 24 h, about 80% of the peak serum concentration is reached. The absorption is sustained with peak serum concentrations reached 72 to 96 h after dosing. The absolute bioavailability is similar to that seen with IFNα-2a. PEG-IFN is found predominantly in the bloodstream and extracellular fluid as seen by the volume of distribution (Vd) at steady-state of 6 to 14 L in humans after intravenous administration. [98].

Ropeginterferon alfa-2b (Ro-PEG-IFN-2b) is a next-generation, monopegylated IFNα designed for extended biological activity and improved tolerability. Its pharmacokinetic (PK) profile is characterized by prolonged absorption, reduced clearance, and an extended half-life compared to conventional IFNα formulations [69]. The peak plasma concentration (Cmax) is reached within 24 to 48 h post-administration, which is significantly delayed compared to non-pegylated IFNα [99]. The absolute bioavailability of Ro-PEG-IFN is high, with minimal variability between doses.

Ro-PEG-IFN demonstrates a volume of distribution (Vd) that is relatively low, indicative of limited tissue penetration and preferential distribution within the extracellular fluid compartment [100].

The metabolism of Ro-PEG-IFN primarily occurs via proteolytic degradation in the reticuloendothelial system and hepatic catabolism. The elimination half-life (t1/2) of Ro-PEG-IFN ranges between 60 and 120 h, which is considerably longer than that of standard IFNα (around 4–12 h) and first-generation pegylated IFNs (~40 h) [101].

Ro-PEG-IFN follows nonlinear pharmacokinetics at higher doses, with a dose-dependent increase in exposure but a plateauing effect at supratherapeutic levels. The extended half-life and sustained systemic exposure of Ro-PEG-IFN allow for a more convenient dosing regimen, typically administered every two to four weeks. This reduces the frequency of injections, improving patient adherence and reducing the incidence of adverse effects [69].

To better understand the pharmacological and clinical differences among the main interferon-alpha formulations currently in use, Table 1 provides a comparative overview of INTRON-A (IFN-α-2b)**,** PEGASYS (PegIFN-α-2a)**,** and BESREMI (RopegIFN-α-2b).

## 7. Clinical Studies of IFN Treatment in MPN (All Resumed in Table 2)

### 7.1. IFN in Polycythemia Vera

Polycythemia Vera (PV) is characterized by excessive red blood cell production due to uncontrolled proliferation of hematopoietic stem cells. The pathogenesis is partly shared with ET and is primarily driven by mutations in the JAK2 gene, leading to aberrant activation of the JAK-STAT signaling pathway. PV increases the risk of thrombotic events and can progress to myelofibrosis or acute leukemia. Management focuses on reducing hematocrit levels and minimizing complications through phlebotomy, cytoreductive therapy, and JAK2 inhibitors [102].

**Table 2 cancers-17-03480-t002:** Summary of key clinical trials investigating interferon-based therapies in myeloproliferative neoplasms (MPNs). Abbreviations: ET, essential thrombocythemia; PV, polycythemia vera; MF, myelofibrosis; MPN, myeloproliferative neoplasm; Peg-IFNα, pegylated interferon alfa; Ropeg-IFNα, ropeginterferon alfa-2b; HU, hydroxyurea; RUX, ruxolitinib; CHR, complete hematologic response; MR, molecular response; BM, bone marrow.

Clinical Trials	Study Design	Treatment	Diseases	Results	References
**MPD-RC 112**	Randomized phase III	Peg-INFα2a vs. HU	High-risk ET and PV	After 36 mo: increased CHR and MR in IFN-treated pts	[103]
**DALIAH trial**	Randomized phase III	Peg-INFα2a, Peg-INFα2b or HU	MPN	From 36 to 60 mo: increased CHR and MR	[104]
**PROUD/CONTINUATION-PV Trial**	Randomized phase III	Ropeg-IFNα vs. HU	Early stage PV	After 36 mo: increased CHR and MR in IFN treated pts	[69]
**COMBI-I Study**	Phase II	RUX and Peg-INFα2	Active PV	After 6 mo: CHR in 80% of pts MR at all time points	[105]
**COMBI-II Trial**	Phase II	RUX and Peg-INFα2 combination	Newly diagnosed PV	After 1 mo: CHR After 24 mo: significant MR	[106]
**Low-PV study**	Randomized phase II	Ropeg-INFα and phlebotomy vs. phlebotomy alone	Low-risk PV	After 2 years: higher response rate and improvement of symptoms and blood counts with ropeg-INFα and phlebotomy	[107]
**P1101MF trial**	Phase II	Ropeg-INFα	Prefibrotic primary MF or low- or intermediate-1-risk MF	After 48 weeks: decreased symptoms, reduction in spleen size, CBC improvement, resolution of BM fibrosis (17.4%) and significant MR	[108]
**COMBI I study**	Phase II	Peg-INFα2 and RUX	MF	Rapid clinical, hematologic, and histological responses with MR	[105]
**RUXOPEG**	Phase I/II	Peg-INFα2 and RUX	MF	Rapid clinical, hematologic, and histological responses with MR	[68]
**SURPASS-ET** (NCT04285086)	Randomized Phase III	Ropeg-INFα2b vs. anagrelide	ET, intolerant or resistant to HU	Ongoing	*unpublished*
**EXCEED-ET** (NCT05482971)	Phase II	Ropeg-INFα2b	ET, intolerant or resistant to HU	Ongoing	*unpublished*

The efficacy of IFNα in the treatment of PV was first demonstrated in the 1980s, making IFNα the subject of many clinical trials that grew more numerous with the introduction of pegylated formulations that significantly improved IFNα stability and toxicity [109].

PV and ET patients refractory or intolerant to hydroxyurea (HU) have shown major responses to pegylated IFNα treatment [110].

Multiple early studies demonstrated that IFNα treatment decreased mutant JAK2 allele burden of the bone marrow and reduced marrow fibrosis leading to long-term remissions, with normalization of blood counts, alleviation of symptom burden, and elimination of splenomegaly in some patients [111].

A phase 3 trial comparing HU to pegylated-IFNα in 168 high-risk treatment naïve PV and ET patients demonstrated similar rates of complete response at 1 year, but improved blood counts and reduced driver mutation frequency between 24 and 36 months with IFNα, highlighting again its disease-modifying activity [103].

In the PROUD CONTINUATION-PV study that compared ropegIFN to hydroxyurea (HU) in a total of 254 patients, long-term follow-up over 6 years showed a clear benefit for patients treated with ropegIFN, including a higher rate of complete hematologic response (CHR, defined by hematocrit below 45% without phlebotomy, a normal white blood cell count, and a normal platelet count) of 54.5% with ropegIFN vs. 34.9% with HU, a much greater reduction in the JAK2 V617F VAF to a median of 8.5% with ropegIFN vs. 50.4% with HU, and a significantly higher probability of event-free survival (including thromboembolic events, disease progression to MF or acute leukemia, or death) in the ropegIFN treatment arm compared with the control treatment group (0.94 vs. 0.82; log-rank test; *p* = 0.04) [69].

Risk events, such as thrombotic events and disease progression, occurred in 5 out of 95 patients (5.3%) in the ropegIFN arm compared with 12 out of 74 patients (16.2%) in the control arm.

Finally, in a recent review and metanalysis of IFN treatment in patients with PV, the CHR rate was estimated at 49% [112].

Experts and academic organizations recommend using peg- IFN-α for the treatment of select patients with high-risk PV [113]. Favorable results from phase 3 studies have strengthened its use as a first-line therapy [114].

More recently, it was suggested that a subset of patients with low-risk PV could benefit from ropegIFN therapy, based on the positive results of a study that randomized phlebotomy and aspirin vs. the same strategy plus a fixed low dose (100 mcg/2 wk) of ropegIFN in a total of 127 patients [115]. The addition of ropegIFN to standard treatment resulted in a significantly higher rate of hematocrit control below 45% without disease progression, more CHRs, improvement in symptoms, and reduction in JAK2 V617F VAF after 12 and 24 months of treatment.

Finally, the combination of ruxolitinib and IFNα is under investigation (NCT02742324), and has been shown to improve MPN manifestations even in patients previously intolerant to peg-IFN [105].

### 7.2. IFN in Essential Thrombocythemia

Essential Thrombocythemia (ET) is a chronic MPN primarily characterized by an overproduction of platelets and an increased risk of thrombotic and hemorrhagic events. Although ET generally has a favorable prognosis compared to other MPNs, its management remains challenging due to complications associated with elevated platelet counts and an increased risk of transformation to myelofibrosis or, rarely, acute leukemia.

The pathophysiology of ET has been better understood in recent years, largely due to insights into mutations in genes such as JAK2, CALR, and MPL. The JAK2 V617F mutation, in particular, is present in approximately 50–60% of ET cases, with CALR and MPL mutations accounting for most of the remaining cases [116]. These mutations play a crucial role in aberrant signaling pathways, leading to the excessive production of platelets and cytokines, which contribute to the pro-inflammatory state observed in ET patients.

From a diagnostic standpoint, prefibrotic myelofibrosis (pre-PMF) remains a critical differential diagnosis, particularly in patients with thrombocytosis and bone marrow megakaryocytic hyperplasia. Despite overlapping features, pre-PMF typically shows more pronounced megakaryocytic atypia, increased granulopoiesis, and mild reticulin fibrosis. Since pre-PMF carries a different prognostic and therapeutic implication, its accurate recognition—guided by WHO and ICC criteria—is essential to avoid misclassification and inappropriate treatment escalation [117,118].

IFN, especially peg-IFNα, has been a focal point in the recent literature on ET management, given its unique mechanism and potential disease-modifying effects. Unlike traditional cytoreductive therapies (e.g., hydroxyurea, anagrelide), which primarily aim to reduce platelet counts, IFN has shown the ability to target and potentially reduce mutant clone burden [119]. This effect has been particularly valuable in younger patients and those with a high thrombotic risk, as it may slow disease progression.

A multicenter study by Masarova et al. [120] evaluated peg-IFNα in ET patients and demonstrated significant hematologic and molecular responses, with a reduction in JAK2 V617F and CALR mutation burdens. These findings were consistent with earlier studies, such as the Phase 3 Randomized French MPD-RC 112 trial, which suggested that peg-IFNα might offer a safer alternative to hydroxyurea with fewer concerns about leukemogenicity [109].

A more recent clinical study randomized peg-IFN-α2a vs. HU in 81 treatment-naive patients with high-risk ET [103]. The proportion of patients in CHR at 12 months was comparable, 45% for HU and 44% for peg-IFN. Too few events were observed in the relatively short follow- up period to see any difference between the 2 treatment arms.

A systematic review and meta-analysis analyzed the results reported from 30 clinical studies of IFN (peg and non-peg) in 730 patients with ET [112]. The complete hematologic response (CHR) rate was 59%, and the rate of thromboembolic complications was low, 1.2% per patient-year.

Two studies reported the rate of complete molecular response (CMR) and partial molecular response (PMR) for the JAK2 V617F mutation in patients treated with peg-INF, which were 26% and 90%, respectively [121,122].

It is important to note that the definitions of CMR and PMR may vary between studies. For example, in the study by Quintás-Cardama et al., CMR was defined as undetectable levels of the JAK2 V617F mutation, whereas PMR was defined as a ≥50% reduction in the mutant allelic burden compared to baseline.

One study including only patients with CALR mutations, showing that IFN-α could also target this driver mutation with a decrease in mutant CALR VAF from 41% to 26% after treatment, including in two patients who achieved complete MR [66,123]. Phase 3 studies testing ropegIFN are ongoing in patients with ET resistant or intolerant to other available therapies, including a randomized study with anagrelide (NCT04285086), which will better define the role of this form of IFN in ET [124].

An interesting field of investigation is the use of interferon in juvenile MPN, so called adolescents and young adults (AYA), as the current treatment recommendations are primarily based on older populations, leading to uncertainty about the best therapeutic strategies for younger patients.

A retrospective study on 348 patients (278 with ET, 70 with PV) diagnosed before the age of 25 years was conducted to assess thrombotic risk, progression to secondary MF (sMF), and the impact of cytoreductive treatments. The incidence of thrombosis was 1.9 per 100 patient-years, like older cohorts. The incidence of sMF was 0.7 per 100 patient-years. IFN-α significantly improved myelofibrosis-free survival compared to other treatments (*p* = 0.046), so the authors suggest that INF-α should be considered first-line therapy in AYA patients to reduce long-term disease progression risk [125].

Finally, given the generally indolent course of ET and the variable correlation between platelet count and thrombotic risk, the potential for overtreatment should not be overlooked. The use of IFN in low-risk or asymptomatic patients should be carefully weighed against its side-effect profile and long-term tolerability, reinforcing the need for individualized, risk-adapted therapeutic strategies.

### 7.3. IFN in Myelofibrosis (MF)

Myelofibrosis (MF) is a chronic Ph-negative MPN characterized by bone marrow fibrosis, extramedullary hematopoiesis, risk of progression to acute myeloid leukemia and a complex clinical picture that includes anemia, splenomegaly, and constitutional symptoms. MF can be either primary or develop secondary to PV or ET, and its pathophysiology is primarily driven by clonal hematopoiesis due to mutations in JAK2, CALR, or MPL [126].

Such mutations play a crucial role in the molecular pathology of MF, with JAK2 V617F being the most common mutation, found in about 50–60% of cases.

MF is characterized by the abnormal expression of various pro-inflammatory cytokines, leading to the proliferation of aberrant immature megakaryocytes and granulocytes and reactive bone marrow fibrosis [127,128,129]. Additionally, MF is associated with severe inflammatory responses and immune dysregulation, making therapeutic approaches targeting these processes of particular interest.

IFNα, particularly in its pegylated form, has emerged as a valuable treatment option for early-stage MF and for MF evolving from PV or ET.

Studies indicate that IFN can lead to hematologic improvements, reduce splenomegaly, and alleviate constitutional symptoms, making it a promising treatment, especially for younger patients and those with early disease stages [130].

One significant study by Gisslinger et al. [69] highlighted the long-term efficacy of peg-IFNα-2a in MF, demonstrating a reduction in symptom burden and a favorable impact on molecular remission rates. Furthermore, the Myelofibrosis and Essential Thrombocythemia Study (METS) showed that peg-IFNα not only offers hematologic responses but also reduces mutant allele burden, which may help delay disease progression [131]. These findings underscore the potential of IFN as a disease-modifying agent, although further research is needed to define its long-term impact on survival.

Based on this evidence IFN-α is currently recommended as an alternative to ruxolitinib or hydroxyurea for the treatment of low-risk MF by the National Comprehensive Cancer Network [132].

While IFN offers promising benefits, it has limitations related to side effects, including fatigue, flu-like symptoms, and cytopenias, which can affect patient adherence and tolerance. Nevertheless, ongoing studies are investigating optimized dosing schedules and the use of lower-dose regimens to improve tolerance and manage side effects effectively [69]. There is also interest in exploring IFN in combination with JAK inhibitors, which may provide synergistic effects in targeting both symptoms and disease progression.

Actually 10 studies exploring IFN therapy in MF are registered on clinicaltrials.gov, 2 of them combining IFN with other agents or procedures.

### 7.4. IFN in Chronic Myeloid Leukemia

Relying on available evidence, the role of IFN-α in CML can be regarded as optimistically ambivalent: while it has demonstrated the capacity to induce long-term immunological control of residual leukemic cells and support treatment-free remission in certain patients, its effects are not universally durable or sufficient to ensure consistent molecular remission after discontinuation.

Prior to the advent of tyrosine kinase inhibitors (TKIs), IFN-α was the standard therapy for CML, showing greater efficacy by inducing complete hematological remission (CHR) and prolonging chronic phase compared to HU and busulfan [133].

Since leukemic stem cells in CML can persist despite TKI therapy, as first suggested by Talpaz et al. [134], IFN-α has been explored as a potential adjunct to improve molecular responses and reduce leukemic burden. Several studies have investigated the combination of IFN-α and TKIs to enhance deep molecular response (DMR) and improve treatment free remission (TFR) rates.

The PETALS Trial (EudraCT 2013-004974-82) evaluated the efficacy of nilotinib (NIL) alone versus NIL combined with peg-IFN-α in newly diagnosed Chronic Phase-CML (CP-CML) patients. Results showed that the NIL + Peg-IFN combination achieved higher MR4.5 rates by month 36, with 20% of patients reaching TFR but without significant differences between the two groups, yet with a short follow-up [135].

The DASA-PegIFN study (NCT01872442) evaluated the efficacy and safety of dasatinib combined with Peg-IFNα-2b in newly diagnosed CML patients. Among 79 patients, 61 received Peg-IFN add-on therapy, with 79% and 61% continuing treatment until months 12 and 24, respectively. MR4.5 rates were 25% and 38% at these time points, with 32% and 46% achieving sustained MR4.5 or MR4. Despite frequent grade 3 neutropenia, severe infections were rare [136].

The TIGER-trial (NCT01657604), a randomized phase III study, evaluated the efficacy and tolerability of nilotinib versus nilotinib + pegylated IFN-a (30–50 μg/week according to tolerability, initiated after ≥6 weeks nilotinib monotherapy) combination therapy for 2 years, followed by continuation of nilotinib in the standard arm versus IFN-a maintenance in the investigational arm. The study accrued 692 patients with newly diagnosed CML-CP. Achievement of MMR, BCR::ABL1 ≤ 0.1% after >24 months of therapy was the trigger to start the maintenance phase; TFR was offered in patients with ≥12 months persistence of MR4 after >36 months of total therapy. The 24-month MMR and MR4.5 rates were 89% and 49% versus 93% and 64% with nilotinib versus nilotinib + pegylated IFN-a, respectively. In 273 (40%) eligible patients who discontinued therapy (nilotinib, n = 163; nilotinib + pegylated IFN-a, n = 110), the 2-year TFR rates were 53% and 59%, respectively. The 8-year progression-free and overall survival rates were 94% and 92%, and 95% and 94%, respectively. More AEs were encountered with the combination of nilotinib + pegylated IFN-a that impaired its tolerability [137].

The ENDURE Trial (CML-IX, NCT03117816) phase III trial evaluated the role of ropeg-IFN in improving TFR in CML patients who had been in stable deep molecular remission after at least 3 years of TKI therapy [138]. However Ropeg-IFN maintenance following TKI monotherapy discontinuation did not enhance the proportion of patients who sustained MMR.

The long-term responses observed after IFN-α discontinuation are thought to result from the activation of leukemia-specific immunity, particularly through the induction of proteinase-3-specific cytotoxic T-lymphocytes (CTLs), which play a role in leukemia surveillance [139].

A prospective study, the PINNACLE trial, examined the effects of nilotinib + IFN-α in newly diagnosed CP-CML patients by analyzing immune cell dynamics at different time points. From a biological perspective, the PINNACLE trial suggested that interferon primarily exerts its effects by stimulating both innate and adaptive immunity, particularly through the activation of NK and T cells against residual leukemic cells. However, despite this immunological rationale, interferon maintenance after TKI discontinuation did not achieve consistent or durable control of minimal residual disease [140].

A small cohort study by Puzzolo et al. [141] analyzed patients who had discontinued IFN-α over 15 years ago. Also in this case patients exhibited higher levels of T cells producing IFN-γ and TNF-α and an increased population of adaptive NK cells, actively responding to residual leukemic cells.

Overall, these findings highlight the ambivalent role of IFN-α in CML: it can induce immunological effects that support TFR in some patients, yet this alone is insufficient to guarantee durable molecular remission after treatment discontinuation [142,143]. 

### 7.5. IFN in Other Myeloid Malignancies

In Chronic Neutrophilic Leukemia (CNL), IFN-α has been widely employed and remains the only pharmacological agent reported to achieve sustained remissions, as documented in limited case evidence [144,145]. Meyer et al. [146] described two patients with progressive CNL who received IFN-α therapy, both achieving durable clinical responses after 16 and 26 months of treatment, respectively.

In CNL, IFN-α represents the only pharmacologic approach consistently associated with durable remissions, as supported by case reports and small series. Sustained hematologic and clinical responses have been observed with both IFN-α and pegylated interferon, lasting from 16 to over 41 months [147].

Given its safety and efficacy, interferon should be considered a first-line therapy for patients of childbearing age and a second-line or subsequent treatment option for those who have failed prior therapies [148].

Based on a limited published literature [149,150,151,152,153,154,155], IFN-α demonstrated hematologic responses and reversion of organ injury in patients with Chronic eosinophilic leukemia (CEL) and idiopathic hypereosinophilic syndrome (HES). The logic of using IFN-α in these settings is partly extrapolated from its efficacy in other MPNs such as CML, as well as PV and ET, and evidence for cytogenetic remitting activity. Although typically used as a second-line-agent in HES after steroid failure, IFN-α could be used as initial therapy in patients with contraindications or intolerance to steroid therapy [156].

### 7.6. IFN During Pregnancy

Prospective studies suggest that pregnancy in women with MPN is generally well-tolerated, with favorable maternal and fetal outcomes. A detailed preconception discussion between clinician and patient should cover potential risks, reported outcomes, and personalized management approach guided by obstetric history and thrombosis risk. IFN-α is the preferred cytoreductive therapy during pregnancy and is strongly advised for women with a prior history of thrombosis, as well as for those considered low-risk but exhibiting high-risk factors, such as uncontrolled hematocrit levels or recurrent pregnancy loss. Additionally, aspirin (81 mg daily) is recommended for all patients, while low molecular weight heparin (LMWH) should be administered antepartum to those with a history of venous thromboembolism [157].

Nowadays IFN-α remains an attractive option in CML during pregnancy due to its safety; in particular it has a role in controlling the disease when diagnosis occurs during pregnancy, as well as for bridging therapy in maintaining the response previously achieved with TKIs, in patients who discontinue them because of desire of pregnancy when optimal criteria for treatment discontinuation are not met [158].

Because of very low levels in milk and poor oral absorption, it is unlikely that interferon use by a nursing mother presents any serious risk to the breastfed infant (see Drugs and Lactation Database (LactMed)) [159].

## 8. Conclusions and Future Directions

Understanding the mechanisms of action by which interferon plays a key role in the pathogenesis of MPN diseases, together with its pharmacological use as treatment of these disorders, is critical for optimal patients’ care.

In the past few years IFN therapy has faced numerous challenges; nowadays, new formulations of the drug have optimized its tolerance, and new indications have brought interferon back into the therapeutic armamentarium of clinicians for the treatment of MPNs.

Overall, IFN represents a powerful tool to control disease symptoms, achieve hematological response, and even reach molecular remission in MPNs.

## Figures and Tables

**Figure 1 cancers-17-03480-f001:**
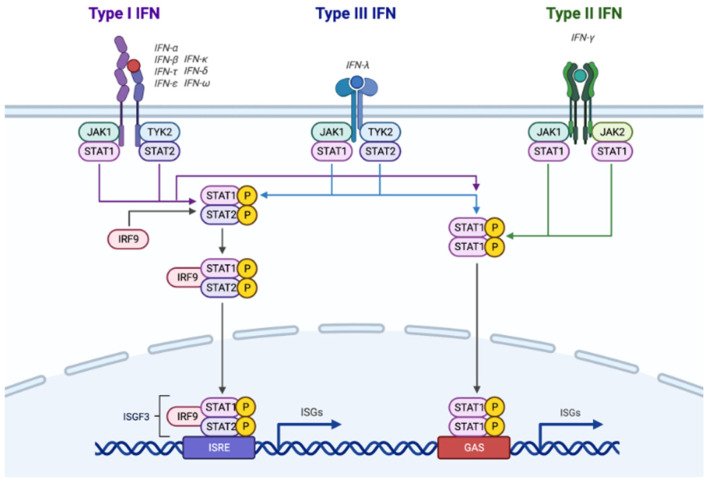
Intracellular signaling pathways activated by interferons (IFNs). Type I, II, and III IFNs bind to their respective receptors (IFNAR1/2, IFNGR1/2, IFNLR1/IL10-R2), triggering the JAK1/2 and TYK2 kinases. This leads to the activation of ISG (Interferon-Stimulated Genes) via key signaling pathways, including MAPK, mTOR, JAK-STAT, and ULK1.

**Table 1 cancers-17-03480-t001:** Comparative table of interferon-alpha formulations. Abbreviations: IFN = Interferon; PegIFN = Pegylated Interferon; SC = Subcutaneous; IM = Intramuscular; MPN = Myeloproliferative Neoplasms; PV = Polycythemia Vera.

Characteristic	INTRON-A (IFN-α-2b)	PEGASYS (PegIFN-α-2a)	BESREMI (RopegIFN-α-2b)
**Type of Interferon**	IFN-α-2b (Standard)	Pegylated IFN-α-2a	Pegylated IFN-α-2b
**Modification**	None	Pegylation (40 kDa, branched)	Pegylation (60 kDa, linear)
**Molecular Weight**	~19 kDa	~40 kDa	~60 kDa
**Half-life**	Short (~4–8 h)	Intermediate (~50 h)	Long (~120–160 h)
**Administration Frequency**	3 times per week	Once per week	Every 2–4 weeks
**Route of Administration**	SC or IM	SC	SC
**Main Indications**	Hepatitis B/C, Melanoma, MPN	Hepatitis B/C, MPN	PV, MPN
**Mechanism of Action**	Activates JAK-STAT pathway, induces antiviral, antiproliferative, and immunomodulatory effects	Prolonged activation of JAK-STAT pathway, enhanced immune modulation	Sustained immune activation, lower systemic toxicity
**Main Adverse Effects**	Flu-like symptoms, cytopenia, liver toxicity	bone marrow suppression, cytopenia	Lower hematological toxicity, better tolerability
**Metabolism and Excretion**	Renal and hepatic clearance	Primarily hepatic metabolism	Hepatic metabolism, prolonged systemic exposure
**Biological Impact**	Strong antiviral and antiproliferative effects	Sustained immune response due to pegylation	Prolonged biological activity with less frequent dosing

## Data Availability

No new data were created or analyzed in this study.

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
