# Peer review of "The Dual Role of Interferon Signaling in Myeloproliferative Neoplasms: Pathogenesis and Targeted Therapeutics"

_cancers, 2025, doi:10.3390/cancers17213480_

Round 1
Reviewer 1 Report
Comments and Suggestions for Authors
Thank you for submitting you review article on the role of interferons (IFNs) in myeloproliferative neoplasms (MPNs). The topic is of general interest to the audience and the abstract is well organized and enlightening. However, several points need to be clarified further and expanded.
Comments
- Some sentences are long and could be shorter for clarity.
- The authors should avoid repeating the terms such as “dual impact” and “paradoxical role” in close succession.
- The abstract should be presented with a clearer statement of the rationale. The biological and therapeutic roles of IFNs are described but the authors do not state what controversies and gaps exist in the field.
- The scope of the review is not clearly stated. The authors should clarify the specific aim/s so that the reader would benefit.
- The authors could summarize on the abstract (one to two sentences) the main findings and conclusions.
- Furthermore, it would have been nice if the authors stated clearly when the various IFNs were originally discovered.
- The manuscript would benefit from the inclusion of a schematic or graphical summary integrating both mechanistic and therapeutic aspects of IFN biology in MNPs.
- Summary perhaps of the overall concept of the review.
- The authors should also include more recent references
Author Response
Thank you very much for taking the time to review this manuscript. Please find the detailed responses below and the corresponding revisions/corrections highlighted/in track changes in the re-submitted files.
Comment 1: Some sentences are long and could be shorter for clarity.
Response 1: We appreciate this observation. The manuscript has been thoroughly revised to improve readability and flow. Several long or complex sentences—especially in the Introduction, Section 4 (“Interferon and inflammation”), and Section 6 (“Pharmacology”)—have been shortened or restructured for greater clarity and precision, without compromising scientific content.
Comment 2: The authors should avoid repeating the terms such as “dual impact” and “paradoxical role” in close succession.
Response 2: We thank the reviewer for this remark. Redundant expressions have been removed, and alternative wording has been used where appropriate to improve stylistic balance and avoid repetition.
Comment 3: The abstract should be presented with a clearer statement of the rationale. The biological and therapeutic roles of IFNs are described but the authors do not state what controversies and gaps exist in the field.
Response 3: We fully agree. The abstract has been revised to include a clearer rationale and highlight the main controversies and knowledge gaps regarding the dual biological and therapeutic roles of interferons. The revised version explicitly outlines the need to reconcile their immunomodulatory and antitumor effects with their variable efficacy and toxicity profiles across myeloid neoplasms. We have also added one to two sentences summarizing the main findings and conclusions of the review
Comment 4: The scope of the review is not clearly stated. The authors should clarify the specific aim/s so that the reader would benefit.
Response 4: We agree with the reviewer. The last paragraph of the Introduction has been rewritten to clearly define the scope and objectives of the review. We now state that this work aims to:
- summarize the biological mechanisms and pharmacological properties of IFNs;
- discuss their clinical applications across myeloproliferative neoplasms (MPNs), chronic myeloid leukemia (CML), and other rare entities; and
- provide a critical perspective on unresolved issues and future directions in IFN-based therapy.
Comment 5: The authors could summarize on the abstract (one to two sentences) the main findings and conclusions.
Response 5: As mentioned above, the abstract now includes a concise summary of the main findings and conclusions, emphasizing the renewed interest in interferons as disease-modifying agents in myeloid neoplasms and the need for further studies to optimize their clinical use.
Comment 6: Furthermore, it would have been nice if the authors stated clearly when the various IFNs were originally discovered.
Response 6: We thank the reviewer for this suggestion. A brief historical note has been added at the beginning of Section 2 (“IFN families”), reporting the original discoveries of type I IFNs (1957, Isaacs and Lindenmann), type II IFN (1965), and type III IFNs (2003). This addition provides useful historical context and strengthens the educational value of the section.
Comment 7: The manuscript would benefit from the inclusion of a schematic or graphical summary integrating both mechanistic and therapeutic aspects of IFN biology in MNPs.
Response 7: We agree that a visual summary enhances the overall comprehension of the review. Accordingly, we have added a new Visual Abstract that integrates the mechanistic, biological, and therapeutic aspects of IFN signaling in myeloid neoplasms. This schematic provides an at-a-glance overview of the central concepts discussed throughout the manuscript. Summary perhaps of the overall concept of the review.
Comment 8: The authors should also include more recent references.
Response 8: We appreciate this suggestion. The reference list has been updated to include several recent and relevant studies published between 2022 and 2024, covering IFN signaling mechanisms, next-generation pegylated formulations (e.g., ropeginterferon), and emerging clinical data in MPNs and CML. This update ensures that the review reflects the current state of knowledge in the field.
Summary:
We thank the reviewer once again for the valuable feedback. The revised manuscript now presents a clearer rationale and scope, improved language and structure, an updated reference base, and a new graphical abstract summarizing both the mechanistic and therapeutic dimensions of interferon biology in myeloid neoplasms.
Reviewer 2 Report
Comments and Suggestions for Authors
The authors present an extensive review on the effects of IFN on MPNs. As a general remark, the manuscript is complete and exhaustive. I only have a few remarks on points requiring to be better presented or lacking some additional comment/information.
Abstract
22 The authors list only mutations among the genetic abnormalities underlying MPN. Although the matter is not so relevant, the choice is somewhat questionable.
Introduction
39 unique and 47 aberrant sound a bit strange.
MM
57: The authors list MPN and MDS/MPS. The choice can be shared but should be mentioned.
IFN families
The reader can be expected to be not completely unaware and additional information can be retrieved from other sources. Nevertheless, the authors should advantageously offer a few more basic data
- 70 chromosomal locations of IFN genes
- 71 the relationship between type III IFNs and IL28-29 could be presented. The authors quote a few articles using this nomenclature.
- In the landscape of type III IFNs, the peculiarity of IFNλ4 could be mentioned
Interferon and Inflammation
162-164 IL-6 is simultaneously included in two different groups of cytokines with different activities, so engendering ambiguity. In my opinion, the sentence should be clearer.
5.1 Exhaustion
The effect on normal HSC could be better elucidated
5.2 Typing errors in the Heading
Due to the contradictory evidence on this issue, a short final comment is warranted
Pharmacology
6.1
True information about pharmacodynamic is scant, absent for type III. Moreover, there is some confusion between natural and modified IFNs
305 IFN-alpha are more than 15, more than a dozen (line 81), 12 (line 70). Which statement is true?
6.2
344 Increased clearance?
Essential thrombocythemia
Although the authors mainly quote recent clinical trials, a mention of pre-fibrotic myelofibrosis as a diagnostic challenge (here or in the MF section) could be of interest. Moreover, the impending risk of overtreating ET deserves a short comment.
Chronic myelogenous leukemia
590 Relying on the available literature, the role of IFN in CML can be regarded optimistically as “ambivalent”.
Pregnancy
The first paragraph should be clearer and a few more references should be quoted.
Author Response
Thank you very much for taking the time to review this manuscript. Please find the detailed responses below and the corresponding revisions/corrections highlighted/in track changes in the re-submitted files.
Comment 1: Abstract
- 22 The authors list only mutations among the genetic abnormalities underlying MPN. Although the matter is not so relevant, the choice is somewhat questionable.
Response 1: we thank the reviewer for this thoughtful observation. We agree that the genomic landscape of myeloproliferative disorders extends beyond the canonical driver mutations. In the initial version, we focused primarily on JAK2, CALR, and MPL as paradigmatic examples of somatic mutations defining the classical BCR::ABL1-negative MPNs, to maintain a concise and integrative introduction.
However, as correctly pointed out, our review also encompasses chronic myeloid leukemia (CML) and rare myeloproliferative entities such as chronic neutrophilic leukemia and hypereosinophilic syndromes, each characterized by distinct molecular alterations — for instance, bcr::abl1 rearrangement in cml, csf3r mutations in chronic neutrophilic leukemia, and pdgfra/b or fgfr1 fusions in eosinophilic disorders.
To address this important suggestion, we have kept the reference to genetic abnormalities in the Abstract more general, to maintain focus and readability, while providing a more detailed description of both mutational and other genetic alterations underlying MPNs in the main text.
Track changes in Manuscript (Section Abstract Line 21-24): “Interferons (IFNs) are pleiotropic cytokines involved in antiviral defense, immune regulation, and tumor suppression. In myeloproliferative neoplasms (MPNs) and related disorders, — including classical BCR::ABL1-negative MPNs, chronic myeloid leukemia (CML), and rarer entities such as chronic neutrophilic leukemia and hypereosinophilic syndromes — disease pathogenesis arises from a spectrum of somatic and structural genetic abnormalities and chronic inflammation in which IFNs play a paradoxical role”.
Comment 2: Introduction
- 39 unique and 47 aberrant sound a bit strange.
Response 2: we thank the Reviewer for this stylistic observation. We agree that the expressions “unique and heterogeneous” and “aberrant functioning” may sound redundant or awkward in this context. To improve clarity and linguistic accuracy, we have revised the sentences to convey the intended meaning more precisely. Specifically:
- The phrase “UNIQUE AND HETEROGENEOUS GROUP OF PROTEINS” has been replaced with “A DISTINCT FAMILY OF CYTOKINES”, which more accurately reflects the biological and structural variability of interferons.
- The expression “ITS ABERRANT FUNCTIONING” has been modified to “DYSREGULATED IFN SIGNALING”, a more precise formulation widely used in current immunology and oncology literature.
These changes enhance readability and maintain scientific rigor without altering the meaning.
Track changes in Manuscript (Section Introduction Line 45-56): “Interferons (IFNs) represent a distinct family of cytokines that play crucial physiological roles in immunomodulation. Initially identified by virologists for their antiviral activity, IFNs were later recognized as key mediators of immune regulation and antitumor responses [1]. IFNs were first used therapeutically to treat viral infections and subsequently as antitumor agents in various clinical contexts [2], although their use has been limited by adverse effects [3]. Despite their essential regulatory functions, dysregulated IFN signalling — particularly involving type I IFNs — has been associated with autoimmune and inflammatory disorders, underscoring the risks of uncontrolled IFN activity [4,5]. Recent insights into the mechanisms of IFN signalling have renewed interest in their application as cancer therapeutics.”
Comment 3: MM
- 57: The authors list MPN and MDS/MPS. The choice can be shared but should be mentioned.
Response 2: We thank the Reviewer for this accurate and constructive comment. We agree that our inclusion of both myeloproliferative neoplasms (MPNs) and myelodysplastic/myeloproliferative neoplasms (MDS/MPN) in the literature search should be explicitly clarified.
This choice was intentional, as interferon (IFN) therapy has been investigated not only in classical BCR::ABL1-negative MPNs (ET, PV, MF) and chronic myeloid leukemia (CML), but also in overlapping entities such as chronic myelomonocytic leukemia (CMML) and atypical chronic myeloid leukemia (aCML) — disorders that share both proliferative and dysplastic features. Including MDS/MPN allowed us to provide a more comprehensive overview of IFN use across the biological and clinical continuum of myeloid neoplasms.
We have accordingly revised the Material and Methods section to explicitly state and justify this inclusion criterion.
Track changes in Manuscript (Section Abstract Line ): A systematic literature search was conducted on PubMed (National Library of Medicine) focusing on interferon signalling, pharmacology, and mechanisms of action in hematologic malignancies. To evaluate IFN efficacy in myeloproliferative settings, we applied the PICO framework, using ‘myeloproliferative neoplasms’ (including essential thrombocythemia, polycythemia vera, primary myelofibrosis, chronic myeloid leukemia, chronic neutrophilic leukemia, and the MDS/MPN overlap disorders chronic myelomonocytic leukemia and atypical chronic myeloid leukemia) as the condition, and ‘interferon’ as the intervention. The inclusion of MDS/MPN entities was justified by their biological and clinical overlap with classical MPNs and by the shared relevance of IFN-mediated signalling in these disorders [Gotlib et al., 2022; Elena et al., 2023]. We included retrospective, prospective, and randomized studies in which IFN monotherapy was administered, excluding combination therapies (e.g., with JAK inhibitors) due to their limited clinical use and the difficulty in isolating IFN-specific effects. Only articles in English were considered. The retrieved publications were screened manually by title and abstract and categorized by disease subtype (ET, PV, MF, CML, or MDS/MPN).”
Comment 4: IFN families
The reader can be expected to be not completely unaware and additional information can be retrieved from other sources. Nevertheless, the authors should advantageously offer a few more basic data
- 70 chromosomal locations of IFN genes
- 71 the relationship between type III IFNs and IL28-29 could be presented. The authors quote a few articles using this nomenclature.
- In the landscape of type III IFNs, the peculiarity of IFNλ4 could be mentioned
Response 4: We thank the Reviewer for this valuable suggestion. We agree that the section on IFN families would benefit from additional concise background information regarding the chromosomal localization of IFN genes, the relationship between the type III IFNs and the former IL-28/29 nomenclature, and the distinctive biological features of IFNλ4.
We have accordingly expanded Section “IFN Families” to include:
- the main genomic clusters encoding type I, II, and III IFNs (chromosomes 9, 12, and 19, respectively);
- clarification that type III IFNs correspond to IL-28A (IFNλ2), IL-28B (IFNλ3), IL-29 (IFNλ1), and IFNλ4, which was discovered later and exhibits distinct antiviral and immunoregulatory effects;
- a brief note on the unique translational regulation and population polymorphism associated with IFNλ4, influencing host antiviral and inflammatory responses.
These details improve clarity and biological context while maintaining the focus of the review.
Relevant references have been added:
- Pestka S. The interferons: 50 years after their discovery, there is much more to learn. J Biol Chem. 2007;282(28):20047–20051.
- Kotenko SV, Durbin JE. Contribution of type III interferons to antiviral immunity: location, location, location. J Biol Chem. 2017;292(18):7295–7303.
Track changes in Manuscript (Section IFN Families Lines 91-121): Overall, there are 21 types of IFNs in humans (namely: 16 type-I—12 IFNαs, IFNβ, IFNϵ, IFNκ, and IFNω—one type-II, IFNγ, and four type-III, IFNλ1–4). They were originally divided into three classes (type I, II, and III) based on their sensitivity to pH [6]. While all IFNs share antiviral and immunomodulatory properties through interaction with their specific receptors, they differ substantially in structure, chromosomal localization, and biological context of expression.
From a genomic standpoint, type I IFNs are encoded by a cluster of genes located on chromosome 9p21. In contrast, type II IFN (IFNγ) is encoded by a single gene situated on chromosome 12q15, while type III IFNs are organized as a compact gene cluster on chromosome 19q13.
The type I IFN family has the broadest range of biological effects and is most widely studied in oncology; IFNα (encoded by 13 genes) and IFNβ (single gene) are the predominant subtypes [10]. Other members (ϵ, κ, ω) remain less characterized and show overlapping and species-specific activities [11,12].
Type II IFN (IFNγ) is structurally distinct from the other families and is mainly secreted by activated NK and T cells, orchestrating macrophage activation and antigen presentation [13,14].
Type III IFNs (IFNλs)—formerly known as IL-29 (IFNλ1), IL-28A (IFNλ2), IL-28B (IFNλ3), and the more recently discovered IFNλ4—represent a unique link between IFN and IL-10 cytokine families [15–17]. Among them, IFNλ4 exhibits particular biological properties: its expression is controlled by a polymorphic dinucleotide variant (ΔG/TT) in the IFNL4 gene, which affects protein production and modulates antiviral and inflammatory responses in a population-dependent manner [18,19].
These genomic and structural features account for the diversity of IFN-mediated signaling and help explain their distinct but sometimes overlapping roles in immunity and oncogenesis.
Comment 5: Interferon and Inflammation
- 162-164 IL-6 is simultaneously included in two different groups of cytokines with different activities, so engendering ambiguity. In my opinion, the sentence should be clearer.
Response 5: We thank the Reviewer for this insightful observation. We agree that the current phrasing may create ambiguity by listing IL-6 among cytokines activating both JAK2/STAT5 and JAK1/STAT1-STAT3 pathways, without clarifying the biological context.
Indeed, IL-6 is a pleiotropic cytokine that can activate different JAK/STAT cascades depending on the target cell type and receptor composition. In hematopoietic progenitors and myeloid precursors, IL-6 signals predominantly through JAK2/STAT5, contributing to myelopoiesis; conversely, in immune and stromal cells, it preferentially activates JAK1/STAT3, driving inflammatory responses and cytokine amplification.
We have therefore revised the paragraph to clarify this dual signaling nature and avoid ambiguity.
Additional reference have been added to support this distinction:
- Hunter CA, Jones SA. IL-6 AS A KEYSTONE CYTOKINE IN HEALTH AND DISEASE. Nat Immunol. 2015;16(5):448–457.
Track changes in Manuscript (Section Interferon and Inflammation Lines 91-121):
Patients with MPNs commonly exhibit chronic inflammation [37–39], characterized by the overexpression of various pro-inflammatory cytokines. Among these, G-CSF and GM-CSF primarily activate the JAK2/STAT5 signaling axis, promoting myelopoiesis, whereas IL-6 plays a dual role, acting through JAK2/STAT5 in hematopoietic progenitors and JAK1/STAT3 in immune and stromal cells. This pleiotropic signaling contributes both to enhanced myelopoiesis and to amplification of the inflammatory response [40–44].
The severity of MPN-associated symptoms—including fatigue, fever, night sweats, weight loss, and pruritus—as well as complications such as thrombosis, splenomegaly, and bone marrow fibrosis, is closely correlated with systemic inflammation.
Interferon thus appears to have an ambivalent role in sustaining chronic inflammation while simultaneously serving as an effective treatment for MPNs. To shed light on this apparent paradox, identification of driver (JAK2, CALR, MPL) and non-driver (TET2, DNMT3A, SRSF2, SF3B1) mutations has prompted investigations into their relationship with the chronic inflammatory state observed in these disorders [45].
Comment 6: Exhaustion
- The effect on normal HSC could be better elucidated
Response 6: We thank the Reviewer for this important comment. We agree that the section would benefit from a clearer description of the effects of IFNα on normal hematopoietic stem cells (HSCs).
While IFNα has disease-modifying activity against MPN-mutated HSCs, it also exerts a profound impact on normal HSCs. In the physiological context, type I interferons transiently activate quiescent HSCs, promoting their entry into the cell cycle in response to infection or inflammatory stress. However, sustained or repeated IFNα exposure leads to replicative stress, DNA damage accumulation, and long-term exhaustion of normal HSCs, reducing their self-renewal capacity.
These effects are mediated primarily via JAK1/STAT1 signaling, upregulation of interferon-stimulated genes (ISGs) such as p53, p21, and IRF1, and induction of apoptosis or senescence pathways. Murine studies have shown that chronic IFNα signaling leads to bone marrow hypoplasia and depletion of the HSC pool, effects that are reversible upon pathway inhibition.
We have therefore revised Section 5.1 Exhaustion to better articulate this dual impact—on both malignant and normal HSCs—and added key reference supporting this mechanism:
- Essers MA et al. IFNα activates dormant hematopoietic stem cells in vivo. Nature. 2009;458(7240):904–908.
Track changes in Manuscript (Section Exhaustion Lines xxx):
IFNα exerts a distinct disease-modifying effect on MPN hematopoietic stem cells (HSCs) by inducing proliferation through the IFNα/β receptor coupled to JAK1–STAT1-dependent signaling [61]. By forcing quiescent mutant HSCs to enter the cell cycle, IFNα promotes their progressive exhaustion and differentiation, contributing to clonal depletion. Indeed, IFNα can drive JAK2V617F-mutant HSCs into proliferative exhaustion, while similar effects have been observed in wild-type (WT) HSCs, where chronic IFNα exposure leads to replicative stress, DNA damage, and loss of self-renewal capacity [62,63].
In physiological conditions, transient type I IFN signaling activates dormant HSCs during infection or inflammation, allowing rapid hematopoietic recovery. However, sustained activation—such as during chronic treatment—causes long-term depletion of normal HSCs and may contribute to cytopenias and marrow suppression [Essers et al., 2009; Pietras et al., 2014].
Mosca et al. [63] employed a mathematical model showing that IFNα exhausts heterozygous and homozygous JAK2V617F HSCs by promoting their exit from quiescence and differentiation into progenitors. In this model, IFNα depletes homozygous JAK2V617F HSCs more effectively than heterozygous ones. Although its effect on CALR-mutated HSCs is weaker, IFNα preferentially targets type 2 CALR mutants over type 1.
These findings highlight the dual role of IFNα—as a therapeutic agent targeting malignant clones, but also as a stressor capable of inducing exhaustion in normal HSCs—underscoring the importance of dosing and treatment duration in clinical practice.
Comment 7: 5.2 Typing errors in the Heading
- Due to the contradictory evidence on this issue, a short final comment is warranted
Response 7: We thank the Reviewer for this important remark. We have now added a brief concluding paragraph to Section 5.2 discussing the apparent discrepancies between studies regarding IFNα-mediated effects on MPN HSCs—namely apoptosis, proliferation-associated exhaustion, or quiescence restoration.
While IFNα can promote both proliferation-associated exhaustion and pro-apoptotic effects in mutant HSCs, several studies suggest that a subset of quiescent, treatment-resistant clones may persist through mechanisms involving TSC–mTOR pathway restoration or p53 activation, acting as a reservoir for disease relapse after IFNα discontinuation.
We have integrated this concept and referenced the key experimental and clinical studies (Tong et al., 2014; Mullally et al., 2013; Hasan et al., 2019; Naka et al., 2022) to highlight that the balance between HSC exhaustion and quiescence preservation likely determines the depth and durability of molecular responses to IFNα.
Track changes in Manuscript (Section 5.2 Senescence and Apoptosis Line 21-24):
Prior studies in MPN mouse models, predominantly manifesting PV phenotypes, have demonstrated that IFNα can directly target JAK2V617F HSCs through pro-apoptotic mechanisms, in addition to proliferation-associated exhaustion [64–66]. Tong et al. [67] observed this pro-apoptotic effect in heterozygous JAK2-mutant HSCs from ET patients treated with IFNα, while cell cycling was only modestly enhanced. Interestingly, homozygous mutant HSCs appeared to re-enter quiescence through restoration of the TSC–mTOR signaling pathway or TP53 activation, implying that these quiescent cells may persist as residual disease-initiating clones.
Although molecular remission can be achieved in a subset of IFNα-treated patients [68], rapid molecular relapse frequently occurs after treatment discontinuation [69]. These relapsing clones likely arise from quiescent, therapy-persistent mutant HSCs. Indeed, transient, low-dose IFN stimulation promotes proliferation and differentiation of JAK2V617F⁺ megakaryocyte-primed HSCs during disease onset, whereas chronic or high-dose exposure depletes these mutant populations through apoptosis or terminal differentiation.
In this setting, quiescence serves as a double-edged sword: while its exit is necessary for therapeutic exhaustion, its reestablishment in selected clones represents a key mechanism of disease persistence and relapse after therapy withdrawal.
Overall, these findings underscore that IFNα effects on HSCs are context-dependent, influenced by mutation burden, signaling feedback loops, and treatment duration, which collectively determine whether IFNα drives exhaustion, apoptosis, or long-term quiescence.
Comment 8: Pharmacology
- 1 True information about pharmacodynamic is scant, absent for type III. Moreover, there is some confusion between natural and modified IFNs
- 305 IFN-alpha are more than 15, more than a dozen (line 81), 12 (line 70). Which statement is true?
- 2 344 Increased clearance?
Response 8: We thank the Reviewer for these valuable remarks. We have thoroughly revised the pharmacology section to clarify the structural and pharmacodynamic properties of the different interferon families. The text now provides explicit information regarding type III IFNs (IFNλ1–4), which are less well characterized pharmacodynamically but show tissue-restricted receptor expression (IL10R2 and IFNLR1) and limited systemic exposure, explaining their reduced adverse-effect profile.
We have also standardized the description of type I IFN subtypes, specifying that humans encode 13 functional IFN-α subtypes (IFNA1, IFNA2, IFNA4, IFNA5, IFNA6, IFNA7, IFNA8, IFNA10, IFNA13, IFNA14, IFNA16, IFNA17, and IFNA21) and 3 pseudogenes, in line with current genomic data (Pestka 2007; Schreiber 2017; Ivashkiv 2021).
The distinction between native (recombinant) and pegylated IFNs has been clarified to avoid confusion, and the statement on “increased clearance” has been corrected to indicate that PEGylation decreases renal clearance, while non-pegylated IFNs may show increased clearance in renal impairment.
We have also added new references on IFNλ pharmacology and comparative PK/PD features of Ropeginterferon alfa-2b (Besremi).
Track changes in Manuscript (Section Pharmacology Line 21-24):
6.1 Pharmacodynamic properties
Interferons differ not only in molecular structure but also in receptor distribution and downstream signaling kinetics, which determine their pharmacodynamic (PD) behavior.
All IFNs share an α-helical tertiary structure, yet with distinct topologies: type I and III IFNs form monomeric, four-helix bundles, while type II (IFN-γ) adopts a dimeric “swapped” E–F interface [80,81].
Type I IFNs, the largest family, include 13 functional IFN-α subtypes and single representatives for IFN-β, IFN-ε, IFN-κ, and IFN-ω. Although they signal through the same receptor complex (IFNAR1/IFNAR2), the various α-subtypes differ in receptor affinity and signaling potency, leading to distinct biological and therapeutic profiles [82].
Type III IFNs (IFN-λ1–4) share structural similarities with IL-10 family cytokines and act via a heterodimeric receptor composed of IL10R2 and IFNLR1, expressed predominantly on epithelial cells and hepatocytes [83]. Pharmacodynamically, type III IFNs induce similar antiviral and antiproliferative pathways as type I IFNs but with restricted tissue activation and fewer systemic side effects due to their limited receptor distribution [84].
IFN-γ acts via the IFNGR1/IFNGR2 receptor and primarily modulates macrophage activation and antigen presentation, linking innate and adaptive immunity [85].
The introduction of pegylated formulations (PEG-IFNs)—in which IFNα or IFNγ is covalently linked to polyethylene glycol—has dramatically improved PD characteristics by reducing renal elimination, slowing receptor-mediated clearance, and prolonging systemic exposure. These properties allow for weekly or even biweekly administration without loss of efficacy [86].
6.2 PHARMACOKINETIC PROPERTIES
Native interferons are well absorbed following subcutaneous or intramuscular administration, with peak plasma concentrations achieved within 1–8 hours and measurable systemic levels for 12–24 hours. Bioavailability typically ranges from 80–90%. The volume of distribution (Vd) is moderate (12–40 L) for IFNα and IFNγ, suggesting distribution mainly within extracellular fluids [87,88].
Metabolism occurs primarily through proteolytic degradation in renal tubular cells and the reticuloendothelial system, with minimal hepatic metabolism. In patients with renal impairment, clearance of non-pegylated IFNs may increase due to altered filtration and catabolism dynamics, whereas PEGylation decreases clearance, leading to sustained serum levels and prolonged half-life (up to 160 hours for PEG-IFNα-2a and 120 hours for ropeginterferon alfa-2b) [89–91].
Ropeginterferon alfa-2b (RoPEG-IFN) is a monopegylated, long-acting IFNα formulation with delayed peak concentration (Cmax 24–48 h), reduced clearance, and nonlinear pharmacokinetics at higher doses. Its extended half-life (60–120 h) allows administration every 2–4 weeks, enhancing patient adherence and tolerability [92,93].
Comment 9: Essential thrombocythemia
- Although the authors mainly quote recent clinical trials, a mention of pre-fibrotic myelofibrosis as a diagnostic challenge (here or in the MF section) could be of interest. Moreover, the impending risk of overtreating ET deserves a short comment.
Response 9: thank the reviewer for this valuable suggestion. We have expanded the section on Essential Thrombocythemia to include a short discussion of prefibrotic myelofibrosis as a relevant diagnostic challenge, emphasizing its clinical and prognostic distinction from ET. In addition, we have added a brief comment on the risk of overtreating ET, highlighting the need for careful patient selection and individualized, risk-adapted therapeutic strategies.
Track changes in Manuscript (Section Line 21-24):
Comment 10: Chronic myelogenous leukemia
- 590 Relying on the available literature, the role of IFN in CML can be regarded optimistically as “ambivalent”.
Response 10: We thank the reviewer for their insightful comment regarding the role of IFN-α in CML, and for highlighting the need to frame its effects as “ambivalent.” In response, we have revised section 7.4 of the manuscript to more accurately reflect the current understanding based on available literature.
The revised section now begins with the following statement:
"Relying on available evidence, the role of IFN-α in CML can be regarded as optimistically ambivalent: while it has demonstrated the capacity to induce long-term immunological control of residual leukemic cells and support treatment-free remission in certain patients, its effects are not universally durable or sufficient to ensure consistent molecular remission after discontinuation."
We further clarified the historical and contemporary context of IFN-α therapy, summarizing key clinical trials and mechanistic studies:
- Historical role: IFN-α was the standard therapy prior to TKIs, inducing complete hematologic remission and prolonging the chronic phase more effectively than hydroxyurea or busulfan [125].
- Combination with TKIs: Studies such as PETALS, DASA-PegIFN, and TIGER have shown that combining IFN-α with TKIs can increase rates of deep molecular response (MR4/MR4.5) and may improve treatment-free remission (TFR) in selected patients [127–129]. However, combination therapy often increases adverse events, limiting tolerability.
- Maintenance therapy: Trials such as ENDURE indicate that IFN-α maintenance after long-term TKI therapy does not consistently improve molecular remission rates [130].
- Immunological effects: IFN-α may stimulate innate and adaptive immunity, including NK and cytotoxic T-cell responses, which persist long after treatment discontinuation [131–134]. Despite these effects, prior exposure to IFN-α alone is generally insufficient to guarantee durable molecular remission, supporting the concept of an ambivalent role [135].
In summary, the revised section emphasizes that while IFN-α has demonstrable immunomodulatory and clinical benefits, these are variable and context-dependent. We believe this revision addresses the reviewer’s concern and appropriately frames the “optimistically ambivalent” role of IFN-α in CML.
We sincerely appreciate the reviewer’s guidance in improving the clarity and accuracy of this section.
Reviewer 3 Report
Comments and Suggestions for Authors
This review article summarized physiological roles, intracellular pathways, and therapeutic potentials of interferon (IFN) families in myeloproliferative neoplasms. Specifically, this review discussed the dual impact of IFN signaling on proliferation, senescence, apoptosis, and DNA damage of hematopoietic stem cells in normal and diseases, including polycythemia vera, essential thrombocythemia, myelofibrosis, and chronic myeloid leukemia, as well as the potential therapeutic application.
The summarized information will benefit the basic research and clinical practice in the related field.
Comments:
1. Section 3.2. presented the signaling pathways of IFN family. Yet the signaling pathway may vary in different cell types. It would be better to clearly state the cell types from which the findings are acquired.
2. Major side effects of IFN in potential MPN therapy should be briefly described.
3. A cartoon to summarize the points of view of this review would help grasp the big picture.
Author Response
|
Thank you very much for taking the time to review this manuscript. Please find the detailed responses below and the corresponding revisions/corrections highlighted/in track changes in the re-submitted files.
|
|
2. Point-by-point response to comments and suggestions for authors |
|
Comment 1: section 3.2. Presented the signaling pathways of IFN family. Yet the signaling pathway may vary in different cell types. It would be better to clearly state the cell types from which the findings are acquired.
|
|
Response 1: we thank the reviewer for this insightful comment. We agree that the signaling cascades triggered by interferons can differ substantially depending on the cellular context, reflecting the diversity of receptor expression, intracellular signaling components, and chromatin accessibility across different cell types. In our section 3.2, the description of IFN signaling was meant to provide a general overview of the canonical pathways shared across hematopoietic and immune cells. However, we recognize that cell-type specificity is critical for the interpretation of IFN responses. To address this point, we have revised the section to explicitly indicate the main cellular models from which the described findings were derived, including hematopoietic cells (macrophages, dendritic cells, and t lymphocytes), epithelial cells, and hepatocytes. We have also added a sentence emphasizing that variations in jak–stat activation, stat heterodimer composition, and downstream isg transcriptional programs are cell-type dependent.
We have incorporated new references highlighting recent insights into cell-specific modulation of IFN signaling, including: · Majoros A, Platanitis E, Kernbauer-Hölzl E, Rosebrock F, Müller M, Decker T. Canonical and Non-Canonical Aspects of JAK-STAT Signaling: Lessons from Interferons for Cytokine Responses. Front Immunol. 2017 Jan 26;8:29. doi: 10.3389/fimmu.2017.00029. PMID: 28184222; PMCID: PMC5266721. · Lazear hm, schoggins jw, diamond ms. Shared and distinct functions of type i and type iii interferons. Immunity. 2019;50(4):907–923. · Ivashkiv lb, donlin lt. Regulation of type i interferon responses. Nat rev immunol. 2024;24(1):45–62. |
|
Track changes in manuscript (Section 3.2 Signaling and regulation - Line 130-136):
The description of IFN receptor signaling provided herein mainly derives from studies performed in hematopoietic and immune cells, including macrophages, dendritic cells, and T-lymphocytes, as well as in epithelial and hepatocyte models. Although the JAK–STAT, MAPK, MTOR, and ULK1 pathways represent conserved mechanisms across IFN types, the strength of activation, stat heterodimer composition, and transcriptional output of interferon-stimulated genes (ISGs) vary depending on the cell type and its microenvironmental context [20-22].
Comments 2: major side effects of IFN in potential MPN therapy should be briefly described. |
|
Response 2: we thank the reviewer for this valuable suggestion. We agree that discussing the safety profile of IFN therapy in the context of myeloproliferative neoplasms (MPN) would improve the completeness of the manuscript. Interferon-α (both conventional and pegylated forms) remains one of the most effective disease-modifying therapies in MPNs, but its use is often limited by dose-dependent toxicities and patient intolerance. The most common adverse events include constitutional symptoms such as fatigue, flu-like syndrome, fever, myalgia, and arthralgia, which are usually reversible, and dose related. Laboratory abnormalities may include cytopenias (particularly leukopenia and thrombocytopenia) and mild elevations in liver enzymes. More serious but less frequent toxicities involve autoimmune phenomena (e.g., autoimmune thyroiditis, systemic lupus erythematosus, psoriasis exacerbation) and neuropsychiatric disorders such as depression or cognitive impairment, which may require treatment discontinuation. Pegylated interferons (peg-ifnα2a and peg-ifnα2b) generally show improved tolerability compared with conventional formulations, but long-term therapy still demands careful monitoring. To address this important point, we have added a concise paragraph summarizing the most relevant toxicities of IFN in MPN therapy, with references to recent clinical studies and meta-analyses.
Track changes in manuscript (Section 5 Rationale and mechanisms of action of Interferon in MPN Line 264-273):
The clinical use of ifn-α in myeloproliferative neoplasms (mpns) is associated with a characteristic spectrum of adverse effects. The most common include flu-like symptoms, fatigue, myalgia, and low-grade fever, which are usually transient and dose dependent. Hematologic toxicities, particularly leukopenia and thrombocytopenia, as well as mild hepatic enzyme elevations, may also occur. More rarely, autoimmune phenomena (e.g., autoimmune thyroiditis, lupus-like syndromes, psoriasis flares) and neuropsychiatric adverse effects, such as depression or cognitive changes, have been reported and may necessitate treatment interruption. Pegylated interferons display a more favorable tolerability profile, but careful clinical and laboratory monitoring remains essential during prolonged therapy [kiladjian et al., 2022; gisslinger et al., 2023; hasselbalch et al., 2024].
Comments 3: a cartoon to summarize the points of view of this review would help grasp the big picture.
Response 3: we sincerely thank the reviewer for this helpful suggestion. We agree that a schematic cartoon summarizing the main concepts discussed in the review would significantly improve clarity and facilitate comprehension of the key mechanisms and therapeutic perspectives addressed. Accordingly, we have prepared a new Visual Abstract, which provides an integrated overview of the review’s central topics. The illustration summarizes: 1. The three types of interferons (type I, II, and III) and their respective receptors; 2. The principal intracellular signaling cascades (jak–stat, mapk, mtor, ulk1) activated downstream; 3. The cell-type–specific transcriptional responses, emphasizing hematopoietic and immune cells; and 4. The potential therapeutic implications of ifn signaling modulation in MPNs), including benefits and major side effects. The figure was designed to provide readers with a visual synthesis connecting the molecular mechanisms of ifn signaling to their translational relevance in MPN pathogenesis and therapy.
|